# Prolonged Hyperoxygenation Treatment Improves Vein Graft Patency and Decreases Macrophage Content in Atherosclerotic Lesions in ApoE3*Leiden Mice

**DOI:** 10.3390/cells9020336

**Published:** 2020-02-01

**Authors:** Laura Parma, Hendrika A. B. Peters, Fabiana Baganha, Judith C. Sluimer, Margreet R. de Vries, Paul H.A. Quax

**Affiliations:** 1Department of surgery; Leiden University Medical Center, 2300 RC Leiden, The Netherlands; l.parma@lumc.nl (L.P.); H.A.B.Peters@lumc.nl (H.A.B.P.); F.Baganha_Carreiras@lumc.nl (F.B.); M.R.de_Vries@lumc.nl (M.R.d.V.); 2Einthoven laboratory for experimental vascular medicine, Leiden University Medical Center, 2300 RC Leiden, The Netherlands; 3Cardiovascular Research Institute Maastricht, Department of Pathology, Maastricht University Medical Center, 6200 MD Maastricht, The Netherlands; judith.sluimer@maastrichtuniversity.nl; 4Centre for Cardiovascular Science, University of Edinburgh, Edinburgh EH16 4SA, UK

**Keywords:** hyperoxygenation, vein graft disease, atherosclerosis, macrophages, vascular biology

## Abstract

Unstable atherosclerotic plaques frequently show plaque angiogenesis which increases the chance of rupture and thrombus formation leading to infarctions. Hypoxia plays a role in angiogenesis and inflammation, two processes involved in the pathogenesis of atherosclerosis. We aim to study the effect of resolution of hypoxia using carbogen gas (95% O_2_, 5% CO_2_) on the remodeling of vein graft accelerated atherosclerotic lesions in ApoE3*Leiden mice which harbor plaque angiogenesis. Single treatment resulted in a drastic decrease of intraplaque hypoxia, without affecting plaque composition. Daily treatment for three weeks resulted in 34.5% increase in vein graft patency and increased lumen size. However, after three weeks intraplaque hypoxia was comparable to the controls, as were the number of neovessels and the degree of intraplaque hemorrhage. To our surprise we found that three weeks of treatment triggered ROS accumulation and subsequent Hif1a induction, paralleled with a reduction in the macrophage content, pointing to an increase in lesion stability. Similar to what we observed in vivo, in vitro induction of ROS in bone marrow derived macrophages lead to increased Hif1a expression and extensive DNA damage and apoptosis. Our study demonstrates that carbogen treatment did improve vein graft patency and plaque stability and reduced intraplaque macrophage accumulation via ROS mediated DNA damage and apoptosis but failed to have long term effects on hypoxia and intraplaque angiogenesis.

## 1. Introduction

The (in)stability of atherosclerotic plaques determines the incidence of major cardiovascular events such as myocardial infarction and stroke [1]. Lack of oxygen within the plaque, or intraplaque hypoxia, has been identified as one of the major contributors to plaque instability [2,3]. It has been detected in advanced human atherosclerotic lesions [4] as well as in murine atherosclerotic lesions [5,6]. 

The intraplaque lack of oxygen is provoked by progressive thickening of the neointimal layer [7] and overconsumption of O_2_ by plaque inflammatory cells [4]. The key regulator of hypoxia is the transcription factor Hif1a [8]. Low oxygen levels, or hypoxia, prevents degradation of Hif1a, promoting its dimerization with the Hif1b subunit. This complex activates the transcription of multiple genes, the most important being Vegfa, that triggers the formation of neovessels in the plaque. Intraplaque neovessels are often immature and therefore leaky, leading to intraplaque hemorrhage, a phenomenon characterized by extravasation of inflammatory cells and red blood cells inside the plaque. Hypoxia also upregulates the expression of transcription factors that cause vascular calcification in vascular smooth muscle cells [9], a characteristic feature of atherosclerosis. Both intraplaque neovessels and vascular calcification are regulated by hypoxia [9] and contribute to plaque instability. The combination of those processes results in a larger plaque which is more unstable and prone to rupture [10,11]. Hif1a was shown to be present in macrophage-rich and foam cell-rich areas and its expression in macrophages was correlated with accelerated atherosclerosis development in LDLR^-/-^ mice [12]. Moreover, it has been shown that hypoxia can influence gene expression in macrophages, leading to an inflammatory response with increased production of pro-inflammatory cytokines [13,14].

Thus, the reoxygenation of the atherosclerotic plaque would be expected to prevent intraplaque hypoxia and atherosclerotic plaque progression. Previously, Marsch et al. showed that plaque reoxygenation in LDLR^-/-^ mice via breathing of the hyperoxic gas carbogen, composed of 95% O_2_ and 5% CO_2_, prevented necrotic core expansion by enhancing efferocytosis [15]. The response of intraplaque angiogenesis to reoxygenation could not be studied in this model, as intraplaque angiogenesis is virtually nonexistent in plaques of LDLR^-/-^ mice. To examine the effect of carbogen treatment on intraplaque angiogenesis we used vein grafts in hypercholesterolemic ApoE3*Leiden mice that do harbor extensive intraplaque angiogenesis, and have been shown to be morphologically similar to rupture-prone plaques in humans. The lesions in this model have the typical characteristics of late stage atherosclerosis, including the presence of foam cells, intraplaque neovascularization, calcification and cholesterol clefts [16] and eventually also occlusion of the graft.

We hypothesized that carbogen gas exposure would reduce hypoxia in vein grafts in the ApoE3* Leiden mice and consequently would reduce intraplaque angiogenesis and increase lesion stability. Thus, we used this model to study plaque reoxygenation and its effect on vein graft remodeling, intraplaque neovascularization, inflammation and vein graft patency. Moreover, since prolonged hyperoxia has the risk of introducing reactive oxygen species [17], we investigated the effect of reactive oxygen species in vitro on bone marrow derived macrophages and in vivo in the atherosclerotic lesions and their effect on the plaque environment.

## 2. Materials and Methods

### 2.1. Mice

This study was performed in compliance with Dutch government guidelines and the Directive 2010/63/EU of the European Parliament. All animal experiments were approved by the animal welfare committee of the Leiden University Medical Center. Male ApoE3*Leiden mice, crossbred in our own colony on a C57BL/background, 8 to 16 weeks old, were fed a diet containing 15% cacao butter, 1% cholesterol and 0.5% cholate (100193, Triple A Trading, Tiel, The Netherlands) for three weeks prior to surgery until sacrifice.

### 2.2. Vein Graft Surgery

Vein graft surgery was performed by donor mice caval vein interpositioning in the carotid artery of recipient mice as previously described [5,18]. Briefly, thoracic caval veins from donor mice were harvested. In recipient mice, the right carotid artery was dissected and cut in the middle. The artery was everted around the cuffs that were placed at both ends of the artery and ligated with 8.0 sutures. The caval vein was sleeved over the two cuffs, and ligated. On the day of surgery and on the day of sacrifice mice were anesthetized with midazolam (5 mg/kg, Roche Diagnostics, Basel, Switzerland), medetomidine (0.5 mg/kg, Orion, Espoo, Finland) and fentanyl (0.05 mg/kg, Janssen Pharmaceutical Beerse, Belgium). The adequacy of the anesthesia was monitored by keeping track of the breathing frequency and the response to toe pinching of the mice. After surgery, mice were antagonized with atipamezol (2.5 mg/kg, Orion Espoo, Finland) and fluminasenil (0.5 mg/kg, Fresenius Kabi, Bad Homburg vor der Ho¨he, Germany). Buprenorphine (0.1 mg/kg, MSD Animal Health, Keniworth, NJ, USA) was given after surgery to relieve pain.

### 2.3. Carbogen Treatment

Acute reoxygenation was investigate in ApoE3*Leiden mice on day 28 after vein graft surgery. Mice were randomized in two groups, a control group (*n* = 13) and a carbogen treated group (*n* = 12) and exposed for 90 min to air (21% O_2_) or carbogen gas (95% O_2_, 5% CO_2_) respectively. Halfway during the treatment, the mice received intraperitoneal injection of hypoxia specific marker pimonidazole (100 mg/kg, hypoxyprobe Omni kit, Hypoxyprobe Inc., Burlington, MA, USA) and anesthesia. Directly after the end of the treatment, mice were sacrificed after 5 min of in vivo perfusion-fixation under anesthesia. Vein grafts were harvested, fixated in 4% formaldehyde, dehydrated and paraffin-embedded for histology.

Chronic reoxygenation was investigated in ApoE3*Leiden mice starting on day 7 after vein graft surgery. The decision for this timepoint was based on our previous finding that intraplaque angiogenesis is detectable in ApoE3*Leiden mice starting from day 14 after vein graft surgery [5]. Mice were randomized based on their plasma cholesterol levels (Roche Diagnostics, kit 1489437, Basel, Switzerland) and body weight in two groups, a control group (*n* = 16) and a carbogen treated group (*n* = 16) and exposed daily for 90 min to air (21% O_2_) or carbogen (95% O_2_, 5% CO_2_) respectively, until the day of sacrifice. On day 28 after surgery, mice received the last treatment and halfway during this last treatment they received intraperitoneal injection of hypoxia specific marker pimonidazole (100 mg/kg, hypoxyprobe Omni kit, Hypoxyprobe Inc., Burlington, MA, USA) and anesthesia. Immediately after the end of the treatment, mice were sacrificed as previously described for the acute reoxygenation experiment.

### 2.4. Histological and Immunohistochemical Assessment of Vein Grafts

Vein graft samples were embedded in paraffin, and sequential cross-sections (5 μm thick) were made throughout the embedded vein grafts. To quantify the vein graft thickening (vessel wall area), MOVAT pentachrome staining was performed. Total size of the vein graft and lumen were measured. Thickening of the vessel wall (measured as intimal thickening + media thickening) was defined as the area between lumen and adventitia and determined by subtracting the luminal area from the total vessel area. The optimal lumen area was calculated by converting the luminal circumference, measured as the luminal perimeter, into luminal area.

Intraplaque angiogenesis was measured as the amount of CD31^+^ vessels in the vessel wall area and intraplaque hemorrhage (IPH) was monitored by the amount of erythrocytes outside the (neo)vessels and scored as either not present, low, moderate or high.

Antibodies directed at alpha smooth muscle cell actin (αSMActin, Sigma, Santa Clara, CA, USA), Mac-3 (BD Pharmingen, Franklin Lakes, NJ, USA), Pimonidazole (mouse IgG1 monoclonal antibody, clone 4.3.11.3, Hypoxyprobe Inc., Burlington, MA, USA), 8OHdG (bs-1278R, Bioss antibodies, Woburn, MA, USA), CD31 (sc-1506-r, Santa Cruz, Dallas, TX, USA), Ter119 (116202, Biolegend, San Diego, CA, USA), Ki67 (ab16667, Abcam, Cambridge, UK) and cleaved caspase 3 (9661-S, Cell SignalingDanvers, MA, USA) were used for immunohistochemical staining. Sirius red staining (80115, Klinipath, Amsterdam, The Netherlands) was performed to quantify the amount of collagen present in the vein grafts. The immuno-positive areas are expressed as a total area or percentage of the lesion area. Stained slides were photographed using microscope photography software (Axiovision, Carl Zeiss Microscopy, White Plains, NY, USA) or Ultrafast Digital Pathology Slide Scanner and associated software (Philips, Cambridge, MA, USA) and image analysis softwares were used to quantify the vein graft intimal hyperplasia and composition (Qwin, Leica, Wetzlar, Germany and Imagej, Bethesda, MD, USA).

### 2.5. RNA Isolation, cDNA Synthesis and qPCR

Total RNA was isolated from 10 (20 µm thick) paraffin sections (at least *n* = 6/group) following the manufacture’s protocol (FFPE RNA isolation kit, Qiagen, Venlo, the Netherlands). cDNA was synthesized using the Superscript IV VILO kit according to the manufacture’s protocol (TermoFisher, Waltham, MA, USA).

Commercially available TaqMan gene expression assays for the housekeeping gene hypoxanthine phosphoribosyl transferase (Hprt) (Mm01545399_m1), and selected genes were used (Applied Biosystems, Foster City, CA, USA); Vegfa (Mm03015193_m1), Hif1a (Mm00468869_m1), Cxcl12 (Mm00445553_m1), Epas1 (Mm01236112_m1), Il6 (Mm00446190_m1), Tnf (Mm00443258_m1) and Ccl2 (Mm00441242_m1). Q-PCRs were performed on the ABI 7500 Fast system (Applied Biosystems). The 2-ΔΔCt method was used to analyze the relative changes in gene expression.

### 2.6. Bone Marrow Derived Macrophages Isolation and In Vitro Experiments

Monocytes were isolated from bone marrow of tibias and femurs of male ApoE3*Leiden mice (*n* = 4) and cultured in RPMI 1640 medium (52400-025, ThermoFisher, GIBCO, Waltham, MA, USA,) supplemented with 25% heat inactivated fetal calf serum (Gibco^®^ by Life Technologies), 100 U/mL Penicillin/Streptomycin (ThermoFisher, GIBCO, Waltham, MA, USA ) and 0.1 mg/mL macrophage colony-stimulating factor (, 14-8983-80, ThermoFisher, E-Bioscience Waltham, MA, USA). After eight days the derived macrophages were seeded in a 12 wells plate for RNA isolation and in a chamber slide for immunocytochemistry (ICC) (NUNC LAB-TEK II, 154534, ThermoFisher, Waltham, MA, USA). 24 h later, when BMM were fully attached, BMM were stimulated with either 200 or 400 µm tert-butylhydroperoxide, t-BHP, (Luperox, 458139, Sigma Aldrich, St. Louis, Missouri, USA) as a ROS mimic for 6 h.

RNA was isolated according to standard protocol using TRIzol^®^ (Ambion^®,^ ThermoFisher,Waltham, MA, USA ) after which sample concentration and purity were examined by nanodrop (Nanodrop Technologies, ThermoFisher, Waltham, MA, USA). Complementary DNA (cDNA) was prepared using the High Capacity cDNA Reverse Transcription Kit (Applied Biosystems, ThermoFisher, Waltham, MA, USA) according to manufacturer’s protocol. For qPCR, commercially available TaqMan gene expression assays for the selected genes were used as explained above.

For ICC cells were fixated in 4% formaldehyde and antibodies directed at Mac-3 (BD Pharmingen, Franklin Lakes, NJ, USA), 8OHdG (bs-1278R, Bioss antibodies, Woburn, MA, USA) and cleaved caspase 3 (9661-S, Cell Signaling, Danvers, MA, USA) were used for immunocytochemical staining. Tile-scans of stained slides were photographed using a fluorescent microscope (Leica AF-6000, Leica, Wetzlar, Germany) and Fiji image analysis software was used to quantify the mean grey value expression of the targets (Imagej, Bethesda, MD, USA).

### 2.7. Statistical Analysis

Results are expressed as mean ± SEM. A 2-tailed Student’s t-test was used to compare individual groups. Non-Gaussian distributed data were analyzed using a Mann-Whitney U test using GraphPad Prism version 6.00 for Windows (GraphPad Software). Probability-values < 0.05 were regarded significant.

## 3. Results

### 3.1. Acute Carbogen Exposure Reduces Intraplaque Hypoxia

To evaluate the effect of acute carbogen treatment on advanced atherosclerotic vein graft lesions, ApoE3*Leiden mice that underwent vein graft surgery were exposed for 90 min to carbogen gas or normal breathing air. Mice exposed to carbogen gas (*n* = 8) showed a significant reduction of intraplaque (IP) hypoxia in the vein graft lesion compared to the air breathing group (*n* = 8) as shown by a 84% decrease in the immuno-area positive for pimonidazole in the lesions of carbogen treated mice compared to the control (*p*-value = 0.027) (Figure 1A–C).

Regarding the aspect of vein graft patency, single 90-min carbogen exposure directly before sacrifice did not affect vein graft patency (Figure 2A), vessel wall area, lumen perimeter, lumen area or optimal lumen area (Figure 2B,C). Furthermore, weight nor cholesterol levels were changed (Appendix A).

The percentage of collagen present in the lesion was comparable between the two groups (Figure 2D) and at a cellular level, the percentage of macrophages (Figure 2E) and SMCs (Figure 2F) were not altered by the acute exposure to carbogen.

### 3.2. Chronic Carbogen Exposure Does not Influence Intraplaque Hypoxia

To evaluate the effect of hyperoxic carbogen treatment on plaque composition and remodeling we performed a chronic carbogen treatment on ApoE3*Leiden mice with advanced atherosclerotic vein graft lesions. Mice were exposed for 90 min daily to carbogen gas (*n* = 13) or normal breathing air (*n* = 12) for 21 days. Neither weight or cholesterol levels were affected by the treatment (Appendix A).

Surprisingly chronic exposure to carbogen gas did not reduce intraplaque hypoxia in the treated group when compared to the air breathing group (Figure 3). In fact, the degree of pimonidazole staining in the vein graft area was not different between the two groups (Figure 3).

### 3.3. Chronic Exposure to Carbogen Plays a Protective Role Against Occlusions

Chronic carbogen treatment resulted in a beneficial effect on vein graft patency, increasing the rate of vein graft patency by 34.5% (Figure 4A). In fact, only 53% of the mice of the control group presented a patent vein graft (Figure 4A), while 87.5% of the mice exposed to carbogen gas had a patent vein graft.

Vessel wall area thickening was not affected by exposure to carbogen gas since no differences could be detected between the two groups when taken only the patent grafts into account (Figure 4B). More importantly lumen size was affected by carbogen gas. In fact, carbogen treated mice presented a significant increase in the lumen perimeter when compared to control (Figure 4C,E, *p*-value = 0.048), and an increase in the optimal lumen area (Figure 4D, *p*-value = 0.067).

### 3.4. Chronic Carbogen Treatment Does Not Have an Effect on Intraplaque Angiogenesis and Intraplaque Hemorrhage

To see whether exposure to carbogen gas had an effect on the hypoxia triggered IP angiogenesis, the amount of CD31^+^ vessels in the vein graft lesions (white arrows in Figure 5A zoom in) was evaluated and no difference in the number of neovessels in the carbogen group was observed when compared to the control group (*p*-value > 0.99).

In addition, when corrected for intimal thickness no differences were observed between the groups (*p*-value = 0.91) (Figure 5A,B). As a measure of the quality of the IP angiogenesis the degree of intraplaque hemorrhage was analyzed (yellow stars in Figure 5A zoom-in) as the amount of Ter119^+^ cells found outside the neovessels and quantified as not present, low, moderate or high, and no differences could be seen when comparing the two groups (Figure 5C).

To determine the effects of hyperoxia on angiogenesis related genes the expression of Hif1a was analyzed. We surprisingly found a significant upregulation of Hif1a mRNA expression in the carbogen treated group when compared to the control (Figure 5D, *p*-value = 0.05), while mRNA expression of Cxcl12, Vegfa and Epas1 were not altered (Figure 5E–G). No differences between control and one-time carbogen treated group were found when analyzing gene expression in vein grafts from the acute carbogen treatment (Appendix A).

### 3.5. Chronic Carbogen Treatment Induces Accumulation of Reactive Oxygen Species and Apoptosis

Although an effect on Hif1a upregulation was observed, surprisingly no effect on angiogenesis could be seen. Therefore, we looked into other mechanisms that could possibly regulate Hif1a. We hypothesized that the mRNA upregulation of Hif1a in the carbogen treated group (Figure 5D) was caused by an accumulation of reactive oxygen species (ROS) induced by the carbogen treatment. ROS is known to be induced by prolonged hyperoxia [17] and to regulate the transcription of different genes involved in hypoxia and in inflammation such as Hif1a and Il6.

Il6 mRNA expression was studied as a representative for ROS induced factors and quantification of its expression showed a trend towards increased expression in the carbogen group of the chronic exposure study when compared to the control group (Figure 6A, *p*-value = 0.09).

Next the presence of ROS was studied in the vein graft lesions by quantifying the amount of ROS–mediated DNA damage, analyzed by 8-hydroxy-2′deoxy-guanosine (8OHdG) immunohistochemical staining. We determined the subcellular location of the staining of 8OHdG. As observed in Figure 6C, a strong 8OHdG positive staining was found in the nuclei of the cells, with an occasional staining outside of the nuclei in the mitochondria, seen as cytoplasmic staining (Figure 6C, right panel). This suggests that the main site of ROS induced DNA damage is nuclear, and not mitochondrial. 8OHdG positive staining could be seen as light blue staining in the nuclei of the cells as a results of co-localized DAPI and 8OHdG staining (Figure 6D zoom-in) and the quantification corrected for the vessel wall area resulted in an increase of DNA damage in the carbogen treated group when compared to the control group (Figure 6B,D), supporting the idea that ROS levels are increased.

ROS is known to induce apoptosis as a consequence of DNA damage. Therefore, the amount of cells positive for cleaved caspase 3 (CC3) in the atherosclerotic vein graft lesions was determined in the carbogen treated and in the control groups (Figure 6E). Due to their high oxygen consumption we hypothesized that macrophages could possibly be the main cell type affected by DNA damage induced by ROS and subsequent apoptosis. As shown in the bottom panel of Figure 6D, macrophages rich areas in the lesions of mice treated with carbogen were found to be strongly positive for CC3 when compared to control (Figure 6E bottom panel).

When looking at the total amount of cells positive for cleaved caspase 3 in the intimal area, an increase in apoptotic cells CC3^+^ in the lesions of mice exposed to carbogen gas was found compared to the air breathing group (Figure 6E, *p*-value = 0.06).

### 3.6. Chronic Carbogen Exposure Reduces Inflammatory Cell Content

The effects of chronic carbogen gas treatment on intraplaque inflammation were studied on macrophages since they produce high amounts of ROS, consume elevate amounts of O_2_ and are known to be hypoxic [6]. Interestingly, the group of mice exposed daily to carbogen for 21 days showed a significant reduction in macrophage content when compared to the control group breathing normal air (*p*-value = 0.0126).

When corrected for the differences in vein graft thickening, the relative percentage of macrophages was significantly decreased in the carbogen exposed group by the 15.2% (Figure 7A, *p*-value = 0.0044).

To study whether the decrease in macrophages was not due to a reduced infiltration of macrophages, nor a reduced proliferation of resident macrophages, local cytokines expression in the vein grafts was studied and the proliferation of macrophages was analyzed.

First, the mRNA expression levels of Ccl2 and Tnf in the vein graft atherosclerotic lesions were examined. The mRNA levels of Ccl2 and Tnf did not differ between the carbogen treated group and the control (Figure 7B,C).

Using a triple IHC staining for Mac-3, Ki-67 and DAPI the amount of proliferating macrophages was determined. As shown in Figure 7D and, there was no difference in the number of proliferative macrophages corrected for the vessel wall thickening (*p*-value = 0.16).

Thus, the data suggest that the reduction of plaque macrophages could be due to enhanced macrophage apoptosis.

### 3.7. Chronic Carbogen Treatment Does Not Affect Plaque Size but Increases Plaque Stability

To evaluate the effect of prolonged carbogen treatment and accumulation of ROS on plaque composition, the amount of collagen (positive collagen area in the total vessel wall) and smooth muscle cells (positive αSMA area in the total vessel wall) was analyzed, two main predictors of plaque stability.

The collagen content in the plaque was not affected by carbogen treatment, and was comparable between the two groups (Figure 8A). Similarly, SMCs content in the carbogen group was not different from the control group (Figure 8B). Interestingly, when calculating the plaque stability index, defined as the amount of collagen and SMCs divided by the vessel wall area, atherosclerotic plaques of the mice daily exposed to carbogen resulted to be more stable than the lesion of the control group (Figure 8C, *p*-value = 0.05).

### 3.8. ROS Increases DNA Damage and Apoptosis in Bone Marrow Derived Macrophages In Vitro

To unravel the molecular and cellular mechanism underlying the observed changes in macrophage content, in particular whether this could be due to hyperoxia induced ROS accumulation, we treated macrophages derived from bone marrow of APOE3*Leiden mice with t-BHP, a known ROS mimic [19]. t-BHP treatment increased the occurrence of DNA damage in BMM as measured by 8-OHdG immunocytochemical staining (Figure 9A), confirming its activity as a ROS mimic and the induction of DNA damage by ROS.

Quantification revealed a 2.2-fold increase in DNA damage in macrophages treated with 200 µm t-BHP (*p*-value = 0.006) and a two-fold increase in DNA damage in macrophages treated with 400 µm t-BHP (*p*-value = 0.006) when compared to control (Figure 9B).

We then evaluated the effect of the ROS mimic t-BHP on the expression of several genes. Similar to changes in expression in vivo, we found that t-BHP-induced ROS caused a significant increase of Hif1a mRNA expression (*p*-value = 0.007 and 0.02 respectively) when compared to control (Figure 9C). Interestingly, we also found that ROS caused an increase in the expression of pro-inflammatory genes Ccl2 and Tnf, but decreased Epas1 expression compared to control (Appendix A)

To assess if ROS ultimately causes apoptosis in cultured BMM, we examined the expression of CC3 and found a significant and dose dependent increase in CC3 expression, thus apoptosis, in t-BHP treated BMM when compared to control (Figure 9F). The group treated with 200 µm t-BHP showed a 10% increase (*p*-value = 0.03) and the group treated with 400 µm t-BHP a 27% increase (*p*-value = 0.01) in CC3 expression when compared to control (Figure 9D). Moreover, we observed a drastic reduction in the total number of cells by 72% and 70% in the groups treated with 200 and 400 µm t-BHP, respectively, when compared to control (Figure 9E, *p*-value = 0.01 for both groups). Combined these data demonstrate that ROS directly affects gene expression in macrophages and causes DNA damage and apoptosis.

## 4. Discussion

The results of the present study show that carbogen treatment in an acute short term setting resulted in a profound reduction of intraplaque hypoxia in murine vein grafts lesions in vivo. Long term treatment with carbogen resulted in a beneficial effect on vein graft patency in ApoE3*Leiden mice, but surprisingly, had no effect on hypoxia, intraplaque angiogenesis and intraplaque hemorrhage. On the other hand, long term carbogen treatment resulted in hyperoxia-induced ROS formation with consequent effects on HIF1a mRNA levels and macrophage apoptosis. A reduction in macrophage content in the vein graft lesions was observed, resulting in less unstable lesions. Moreover, comparable to what was observed in vivo, in vitro induction of ROS using the ROS mimic t-BHP in BMM resulted in a strong increment in DNA damage and apoptosis.

Carbogen inhalation is widely used in the oncological field [20,21]. It has been shown that the time to achieve a maximal increase in tumor oxygenation with carbogen inhalation depends on various factors such as the type of cell involved, the location, and the size of the tumor [22,23]. Moreover, Hou et al. observed an effect of carbogen treatment comparable to what was observed in the present study, both in the short and the long term experiments. Single carbogen inhalation significantly increased tumor oxygenation, while during multiple administrations of carbogen the effect was reduced, indicating that the response to chronic carbogen is not consistent over days [22]. Nevertheless, we showed that prolonged carbogen treatment has a protective role against vein graft occlusions. Vein graft occlusion is a phenomenon often seen after vein grafting in which the vessel lumen is narrowed due to extensive intimal hyperplasia that progress to stenosis and occlusion [24]. This phenomenon is also observed in ApoE3*Leiden mice that undergo vein graft surgery. Besides the reduction in vein graft occlusions, an increase in vein graft patency due to an increase in lumen perimeter and optimal lumen area of the hyperoxic vein grafts was observed, similar to the study by Fowler et al. [25]. In that study carbogen is used in the treatment of central retinal artery occlusion to increase blood oxygen maintaining oxygenation of the retina [25]. This effect of hyperoxygenation on retinal artery remodeling can be related to the effect of carbogen on patency and increase in lumen perimeter and increase in the optimal lumen found in the present study.

We did not observe a reduction in hypoxia nor an effect on intraplaque angiogenesis in the prolonged carbogen study. Furthermore, no changes in local gene expression of Vegfa were observed in the vein grafts, but interestingly Hif1a was upregulated in the prolonged carbogen exposure study and not downregulated as expected. In fact following our initial hypothesis we would have expected a reduction in intraplaque angiogenesis in parallel with a reduction in Hif1a and Vegfa expression. For this reason, we studied other known processes that regulate Hif1a and observed an accumulation of ROS in the carbogen exposed group when compared to the control group. Repeated exposure to hyperoxia is known to be associated at a cellular level with an accumulation of ROS [26,27]. When the exposure is repeated too often, the oxidant insult is no longer compensated by the host’s antioxidant defense mechanisms and therefore cell injury and death ensue [28]. Cell injury induced by ROS comprises lipid peroxidation, protein oxidation and DNA damage [29,30]. We observed an increase in DNA damage measured as an augmented presence of 8OHdG staining in the long term carbogen treated group when compared to the control group, indicating that a daily long term treatment with carbogen gas results in accumulation of ROS that in turns induces DNA damage in the atherosclerotic lesions. Moreover, we also observed an increase in DNA damage in bone marrow macrophages in vitro under ROS stimulation. It is known that DNA damage can be found in the nuclei and in the mitochondria [31,32]. Both in the vein graft lesions in vivo and in the cultured t-BHP treated macrophages in vitro, a strong 8OHdG positive staining in the nuclei of the cells, with an occasional cytoplasmic staining could be seen. The subcellular location of the staining of 8OHdG suggests that the main site of ROS induced DNA damage is nuclear, and not mitochondrial. ROS generated by repeated hyperoxia treatment can alter gene expression by modulating transcription factor activation, like NF-kβ, which then impact downstream targets [33]. It has been shown that hyperoxia also results in nuclear translocation of NF-kβ and NF-kβ activation in several cell types [34]. Our results show that long term carbogen treatment result in Hif1a gene expression upregulation. In addition, in vitro BMM treated with the ROS mimic t-BHP also showed an upregulation of Hif1a gene expression. Interestingly, the transcription of this gene is known to be regulated by NF-kβ transcription factor. In fact, Bonello et al., demonstrated that ROS induced Hif1a transcription via binding of NF-kβ to a specific site in the Hif1a promoter [35]. Those findings could be further investigated in future experiments using antioxidants such as NAC to see whether it can reverse the carbogen treatment.

We showed that the accumulation of ROS in the carbogen treated group caused an increase in apoptosis, accumulated in macrophages rich areas, and resulted in a decrease in the amount of macrophages. Even though we cannot exclude that the association of macrophages with cleaved caspase 3 could be due to efferocytosis of apoptotic cells, macrophage efferocytosis is frequently hampered in atherosclerotic lesions, therefore it is likely that these macrophages are apoptotic. Previously, in contrast with our findings, a strong correlation between macrophage content and hypoxia was shown by Marsch et al. [15]. Moreover, hypoxia potentiates macrophage glycolytic flux in a Hif1a dependent manner [36] in order to fulfill the need of ATP for protein production and migration. Taken together, this points to a high request and high use of O_2_ by plaque macrophages and a consequent high exposure of these inflammatory cells to ROS accumulated during hyperoxia. We demonstrated that ROS causes accumulation of DNA damage and subsequently an increase in apoptosis and cell death in BMM in vitro. The link between ROS induced DNA damage and apoptosis detected in vitro might explain the observed apoptosis in macrophages in vivo. Moreover, a reduction in the number of macrophages is associated with plaque stability and plaque stability is reflected in an increase in vein graft patency as observed in the present study.

Previously Marsch et al. showed that repeated carbogen treatment in LDLR^-/-^ mice lead to reduction in intraplaque hypoxia, necrotic core size and apoptosis [15]. In the present study we showed that repeated carbogen treatment in accelerated vein graft atherosclerotic lesions in ApoE3* Leiden mice resulted in increased apoptosis and unaltered intraplaque hypoxia when compared to controls. Accelerated atherosclerotic lesions in ApoE3*Leiden mice highly resemble human atherosclerotic lesions and, differently from LDLR^-/-^ mice, do present intraplaque angiogenesis. Our results show that although we did not observe reduced intraplaque angiogenesis and IPH daily hyperoxia treatment with carbogen gas in this murine model lead to accumulation of ROS that could not be cleared by anti-oxidant agents and the ROS build-up lead to DNA damage and induced apoptosis. In fact, differently from Marsch et al., who treated mice daily for five days, followed by two days of no carbogen exposure we performed the treatment daily and started our treatment seven days after mice underwent vein graft surgery, when the atherosclerotic lesions already started forming. This starting time point was based on our previous findings [5] in which we found that intraplaque neovascularization in ApoE3*Leiden mice that underwent vein graft surgery is visible 14 days after surgery. Therefore, we were able to study the effect of carbogen treatment on lesion stabilization rather than on lesion formation.

One of the limitations of the current study may be the choice of the model used, the ApoE3*Leiden mice vein grafts. However, since in most mouse models for spontaneous atherosclerosis intraplaque angiogenesis is absent, and the lesions observed in the ApoE3*Leiden mice vein grafts show many features that can also be observed in advanced human lesions, including intraplaque hypoxia, angiogenesis and intraplaque hemorrhage, we believe this model is suitable for the current studies. The fact that the most prominent effects observed relate to hyperoxygenation induced ROS production, macrophage apoptosis and vein graft patency, whereas the experimental set-up was initially designed to identify effects on intraplaque angiogenesis, might indicate another limitation in our study set-up.

Based on the results obtained in the present study we can conclude that although short term carbogen gas treatment leads to a profound reduction in intraplaque hypoxia, the treatment has mixed effects. Despite the beneficial effects of the hyperoxygenation treatment on vein grafts, i.e., improved vein graft patency and a strong trend towards an increased plaque stability index, chronic hyperoxygenation also induced Hif1a mRNA expression, ROS accumulation and apoptosis. That all will harm the vein grafts in the current model under the current conditions. This indicates that in order to define potential therapeutic benefits of hyperoxygenation treatment further research is needed to define optimal conditions for this treatment in vein graft disease.

## Figures and Tables

**Figure 1 cells-09-00336-f001:**
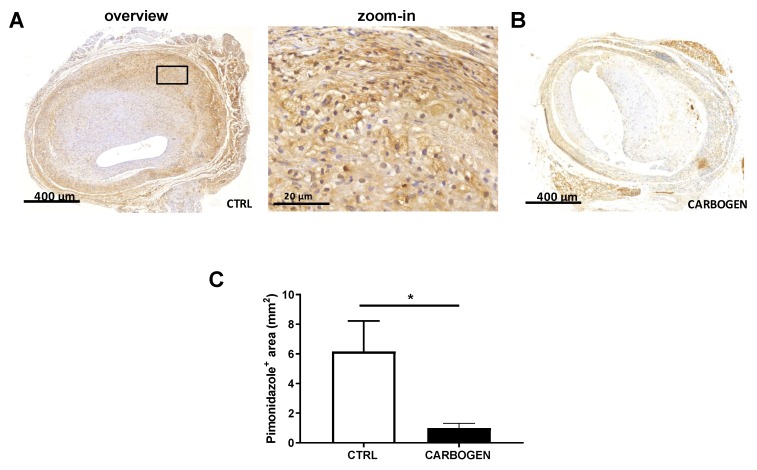
Short term carbogen exposure drastically reduces intraplaque hypoxia. (**A**) Representative pictures of sections from vein graft lesions in ApoE3* Leiden mice stained for pimonidazole in the control group (*n* = 8) and (**B**) one-time carbogen treated group (*n* = 8). (**C**) Quantification of pimonidazole positive area. Data are presented as mean ± SEM. * *p* < 0.05; by two-sided Student’s t test.

**Figure 2 cells-09-00336-f002:**
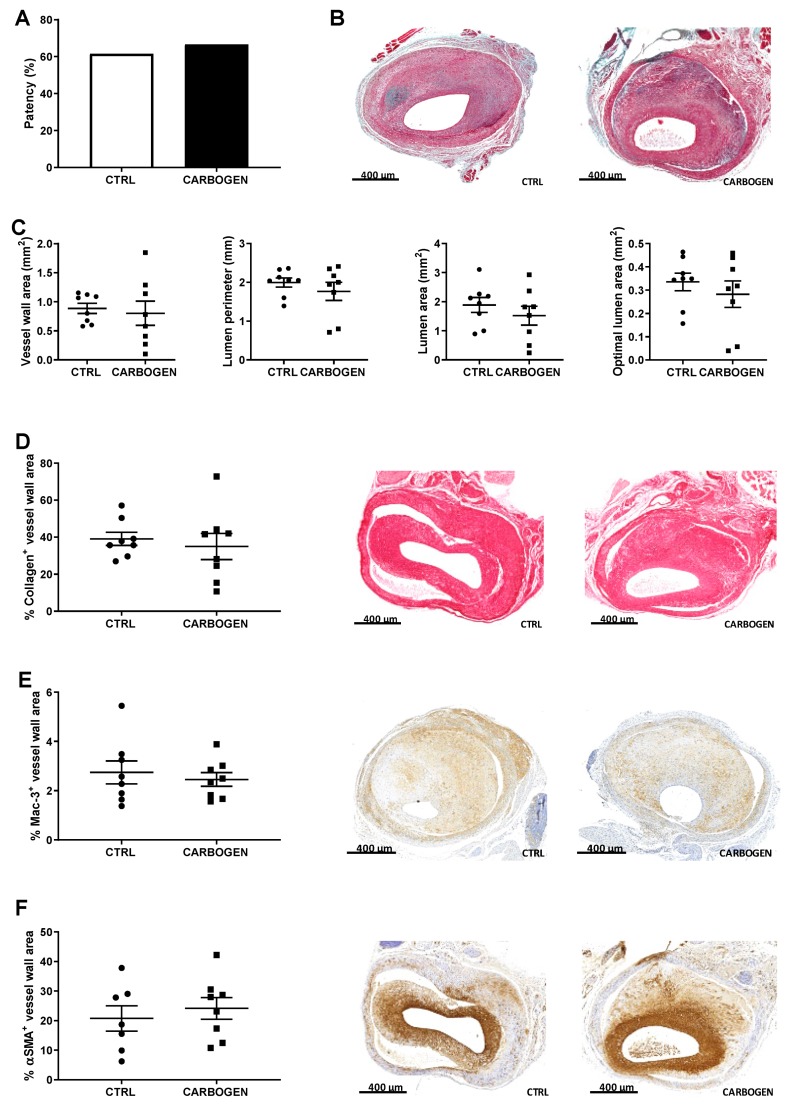
Short term exposure to carbogen gas does not influence plaque size nor composition. (**A**) Quantitative measurement of vein graft patency in ApoE3*Leiden mice from the control and one-time carbogen treated groups. Data are analyzed by Chi-square test (**B**) Representative pictures of MOVAT staining of vein graft sections from control (*n* = 8) and carbogen group (*n* = 8). (**C**) Quantification of vessel wall area, lumen perimeter, lumen area and optimal lumen area. Percentage of positive vessel wall area and representative pictures for (**D**) collagen (*n* = 8 for control and carbogen groups), (**E**) macrophages (*n* = 8 for control and carbogen groups) and (**F**) smooth muscle cells staining (*n* = 8 for control and *n* = 7 for carbogen groups). Data are presented as mean ± SEM.

**Figure 3 cells-09-00336-f003:**
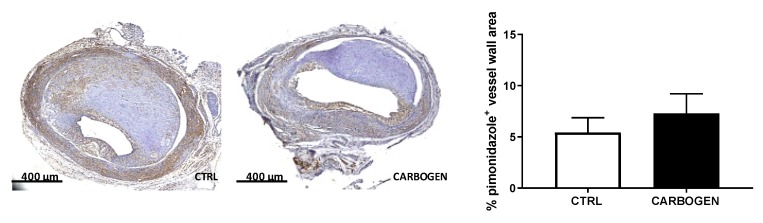
Chronic carbogen treatment does not affect intraplaque hypoxia. Representative pictures of vein graft cross sections stained for pimonidazole in control (*n* = 8) and chronic-treated carbogen groups (*n* = 13) and quantitative measurement of percentage of vessel wall area positive for pimonidazole staining. Data are presented as mean ± SEM.

**Figure 4 cells-09-00336-f004:**
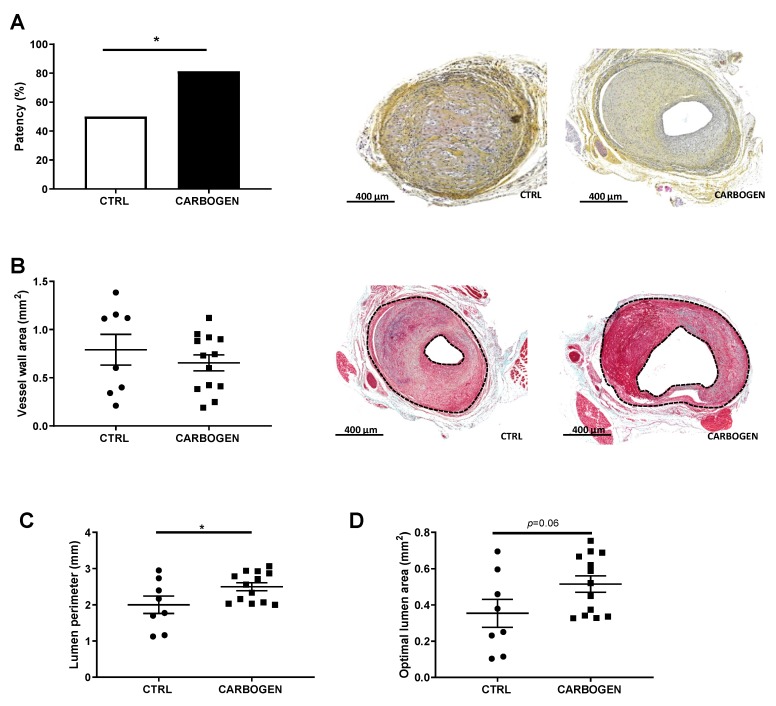
Chronic exposure to carbogen plays a protective role against vein graft occlusions. (**A**) Quantitative measurement of vein graft patency in ApoE3*Leiden mice from the control and prolonged carbogen treated groups. Data are analyzed by Chi-square test. * *p* < 0.05. Representative pictures of non-patent and patent vein grafts in ApoE3* Leiden mice at day 28 after surgery in the right panel. (**B**) Quantitative measurements of vein graft thickening. In the right panel, representative pictures of MOVAT staining in vein grafts from control (*n* = 8) and long term carbogen treated mice (*n* = 13), with vessel wall area as the area between the two dotted lines. (**C**) Quantification of lumen perimeter and (**D**) optimal lumen area. Data are presented as mean ± SEM. * *p* < 0.05; by two-sided Student’s t test.

**Figure 5 cells-09-00336-f005:**
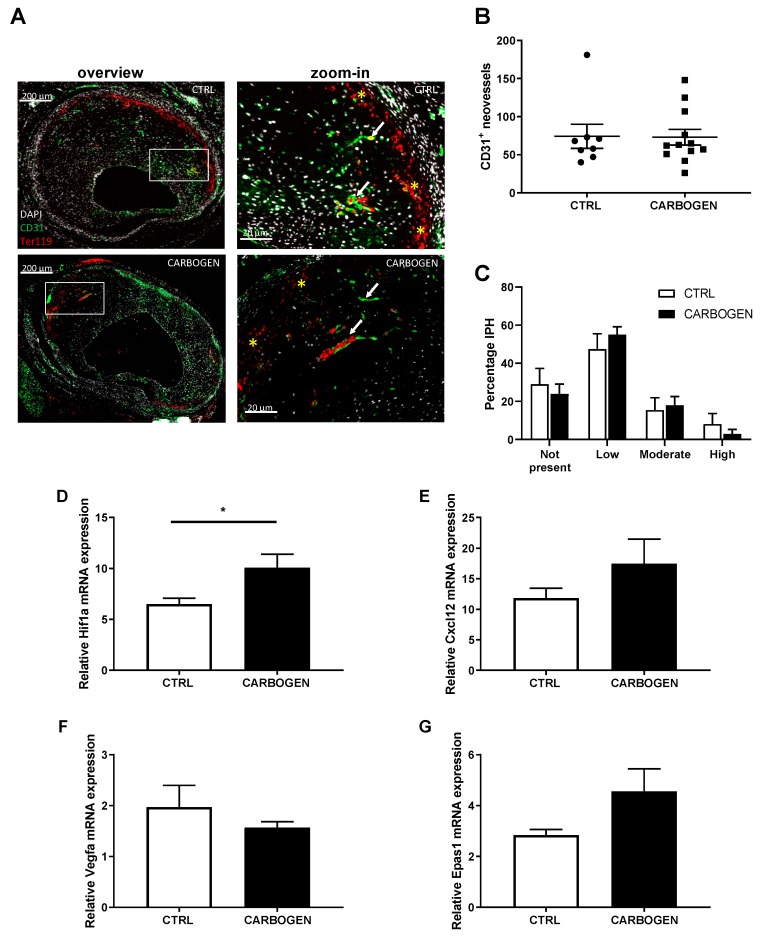
Chronic carbogen treatment does not affect intraplaque neovascularization. (**A**) Representative pictures of vein grafts lesions stained for DAPI (white), CD31 (green) and Ter119 (red). (**B**) Quantification of CD31 positive neovessels in the vessel wall area in the control group (*n* = 8) and in the carbogen treated group (*n* = 12). (**C**) Bar graphs representing the quantitative measurements for IPH in the control and long term carbogen treated groups. IPH was scored as not present, low, moderate or high. Total vessel wall gene expression of (**D**) Hif1a, (**E**) Cxcl12, (**F**) Vegfa and (**G**) Hif2a, relative to Hprt, was measured in the control and long term carbogen treated groups. Data are presented as mean ± SEM. * *p* < 0.05; by two-sided Student’s t test.

**Figure 6 cells-09-00336-f006:**
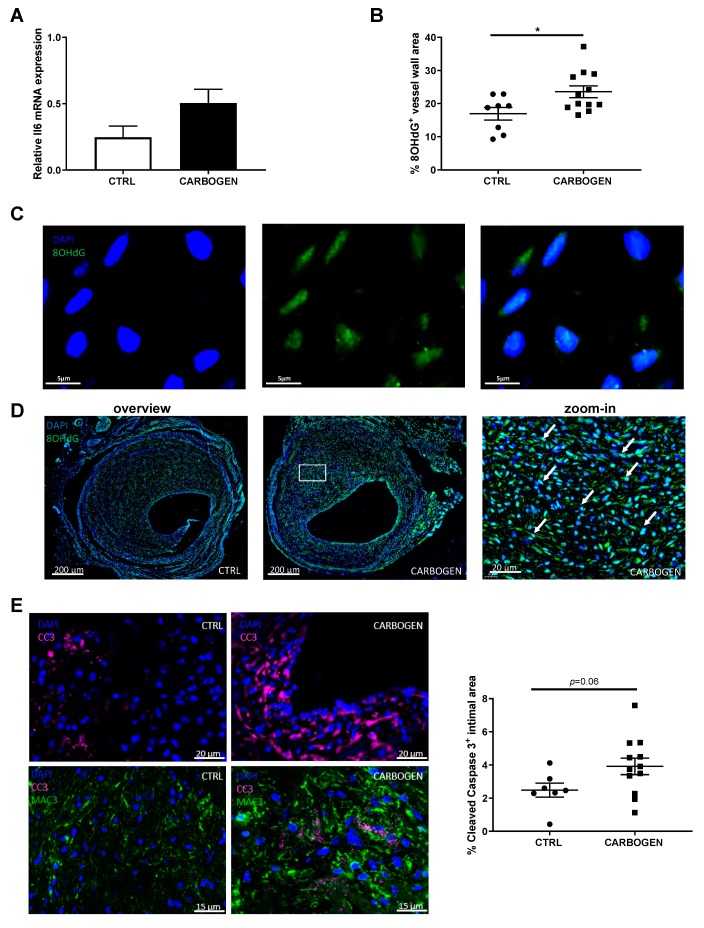
Chronic carbogen treatment induces accumulation of ROS. (**A**) Il6 gene expression relative to Hprt in the total vessel wall of control and chronic carbogen treated groups. (**B**) Quantification of the percentage of vessel wall area positive for 8OHdG. (**C**) Representative pictures of DAPI (in blue, left panel), 8OHdG (in green, central panel) and merged (right panel) staining in the vein graft lesions. (**D**) representative pictures of DAPI (blue) and 8OHdG (green) staining in vein graft lesions from control (*n* = 8) and carbogen treated (*n* = 12) mice. Light blue staining represents nuclei positive for 8OHdG and examples are indicated by white arrows. (**E**) In the top panels, representative pictures of DAPI (blue) and cleaved caspase 3 (CC3, in magenta) staining and in the bottom panels representative pictures of DAPI (blue), CC3 (magenta) and Mac3 (green) staining in control and carbogen groups respectively. In the right panel quantification of percentage of intimal area positive for cleaved caspase 3. Data are presented as mean ± SEM. * *p* < 0.05 by two-sided Student’s t test.

**Figure 7 cells-09-00336-f007:**
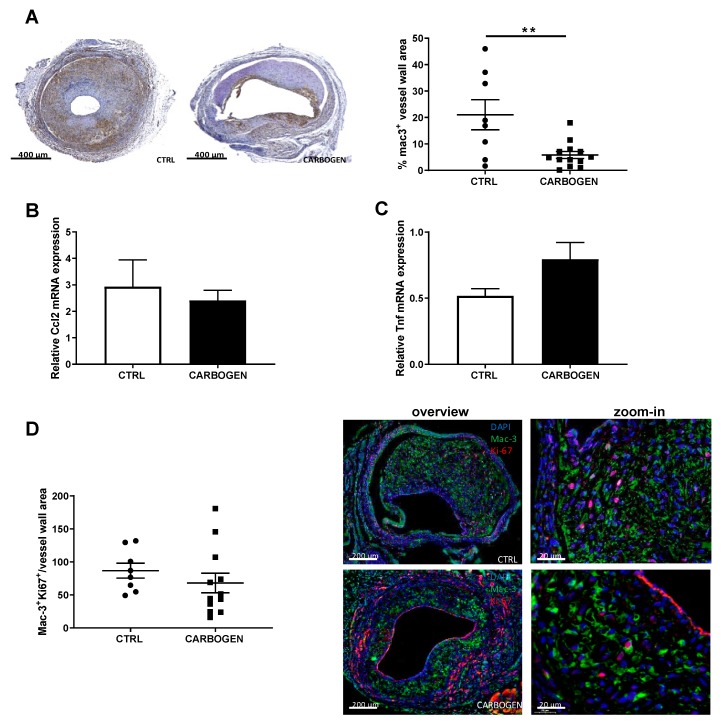
Chronic carbogen treatment reduces macrophages infiltration in the plaque. (**A**) Representative pictures of ApoE3* Leiden mice vein grafts from control (*n* = 8) and chronic carbogen treated (*n* = 13) groups stained for Mac-3. In the right panel quantitative measurements of the percentage of vessel wall area positive for Mac-3. Data are presented as mean ± SEM. ** *p* < 0.01; by two-sided Student’s t test. Total wall gene expression of (**B**) Ccl2 and (**C**) Tnf relative to Hprt. (**D**) Quantification of the number of cells positive for Mac-3 and Ki67 in the vessel wall area of control (*n* = 8) and carbogen treated group (*n* = 13) and representative pictures of the staining with DAPI presented in blue, Mac-3 in green and Ki67 in red. Data are presented as mean ± SEM. ** *p* < 0.01 by 2-sided Student t test.

**Figure 8 cells-09-00336-f008:**
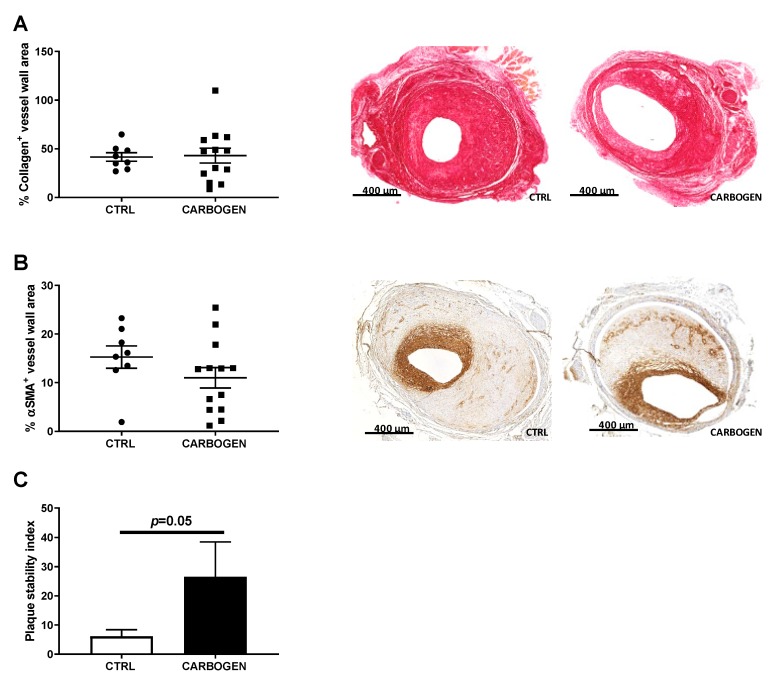
Chronic carbogen treatment does not affect collagen nor smooth muscle cells content in the lesion but increases plaque stability. (**A**) Quantitative measurement vessel wall area positive for collagen in ApoE3*Leiden mice from the control (*n* = 8) and carbogen treated (*n* = 13) groups. In the right panel representative pictures for collagen staining. (**B**) Quantification of percentage of vessel wall area positive for smooth muscle cell actin and representative pictures from the control (*n* = 8) and carbogen treated (*n* = 13) groups. (**C**) Quantification of plaque stability index. Data are presented as mean ± SEM.

**Figure 9 cells-09-00336-f009:**
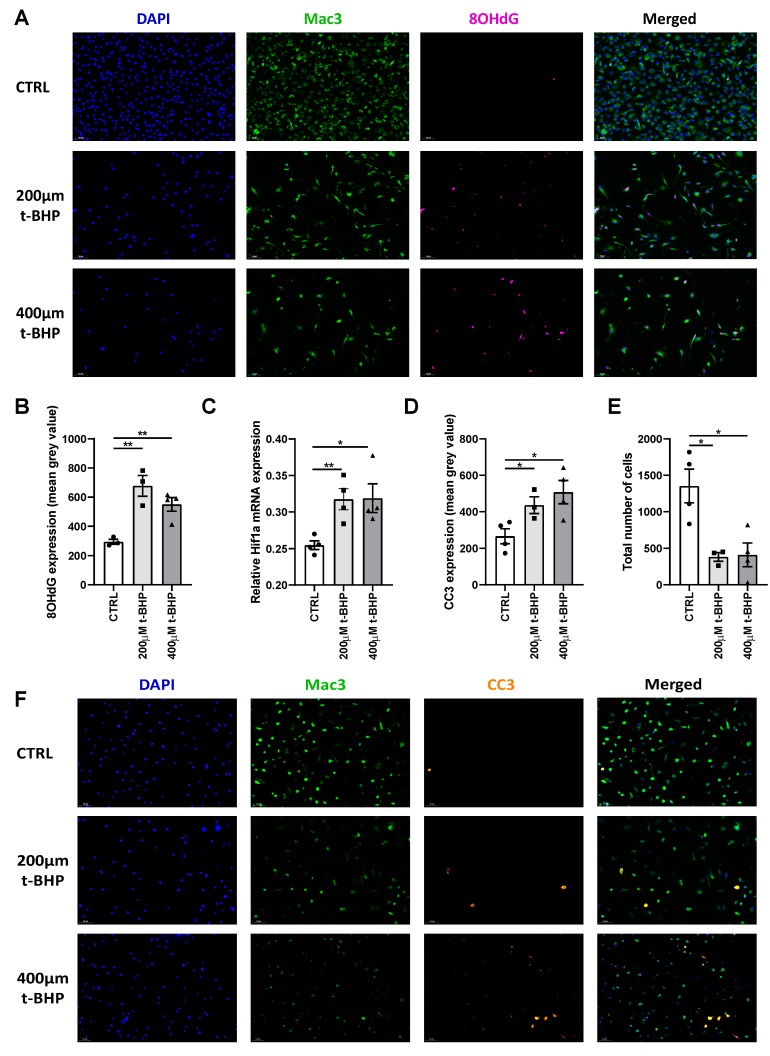
ROS induces DNA damage and apoptosis on in vitro bone marrow macrophages (BMM) (**A**) Representative pictures of CTRL BMM and BMM treated with 200 or 400 µm t-BHP respectively are shown. Examples images stained for DAPI (blue), Mac3 (green) and 8OHdG (magenta) as well as a merged image are shown per each condition tested. (**B**) Quantification of 8OHdG expression as mean intensity is shown. (**C**) Total mRNA expression of Hif1a relative to Hprt. (**D**) Quantification of CC3 expression as mean intensity is shown. (**E**) Quantification of total amount of cells per condition tested expressed as total amount of positive DAPI nuclei. (**F**) Representative pictures of CTRL BMM and BMM treated with 200 or 400 µm t-BHP respectively. Examples images stained for DAPI (blue), Mac3 (green) and cleaved caspase 3 (CC3) (orange) as well as a merged image are shown per each condition tested. Data are presented as mean ± SEM. * *p* < 0.05, ** *p* < 0.01; by two-sided Student’s t test.

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
