# Peer review of "Prolonged Hyperoxygenation Treatment Improves Vein Graft Patency and Decreases Macrophage Content in Atherosclerotic Lesions in ApoE3*Leiden Mice"

_cells, 2020, doi:10.3390/cells9020336_

Round 1

Reviewer 1 Report

There are some typos and grammatical errors that need correcting. Otherwise I am happy with the manuscript in its current form. Thank you.

Reviewer 2 Report

My concerns were addressed, I do not have further comments. 

This manuscript is a resubmission of an earlier submission. The following is a list of the peer review reports and author responses from that submission.

Round 1

Reviewer 1 Report

This manuscript of Parma et al. addresses the effect of carbogen gas on the remodeling of vein graft atherosclerotic lesions in ApoE3*Leiden mice. Vein graft lesions in the ApoE3* Leiden mice exhibit features of late stage human plaques including the presence of foam cells, cholesterol crystals, intraplaque neovascularization and hemorrhage, and calcification. Occlusion of the graft is frequent in this model.  

The authors applied acute and chronic treatment regimens to study the effect of carbogen gas on the remodeling process.

Acute treatment compensated intraplaque hypoxia without affecting plaque composition. Chronic treatment with carbogen resulted in increased vein graft patency and lumen size. Interestingly acute carbogen gas treatment did not improve plaque hypoxia and subsequent neovessel formation and intraplaque hemorrhage. The authors suggested that this is due to the accumulation of ROS and Hif1a induction during the hyperoxic treatment. They showed a reduction in plaque macrophage content due to accelerated apoptosis.

The experiments are well-designed and appropriate to answer the questions.

Comments:

The fluorescent images are too dark on Figure 9. That needs to be changed. The title of Fig. 8 is “Chronic carbogen treatment does not affect plaque composition but increases plaque stability.” is a bit misleading. This title reflects only the results shown in that figure (collagen positive vessel wall area and aSMA positive vessel wall area), but not stand if we look at the whole manuscript (e.g. in the previous figure the authors showed that chronic carbogen treatment reduces inflammatory cell content of the plaque). To avoid confusion, please make the title of figure 8 more specific.

Reviewer 2 Report

This study byt Parma et al investigates the effect of hyperoxygenation on vein graft patency. There are some really lovely histology images in this manuscript. While some of the data is interesting, it is not clear why the particular surgical model used was chosen. It is also unclear why the focus was on the long term hyperoxia treatment, when there were more significant in the short-term treatment. There is a lot of focus on IPH and angiogenesis in the long-term model, but there no changes in these. Why wasn’t the smaller plaque size investigated more thoroughly instead. WHat is the reason for performing this study if one that is quite similar had been done in LDLr KO mice?

Specific comments are below.

The first sentence (line 17) is not correct, angiogenesis may contribute to plaque instability, but not all unstable plaques show angiogenesis. Please change this.

Line 18: “Hypoxia plays a role in angiogenesis and inflammation and subsequently in the pathogenesis of atherosclerosis” This sentence is not quite correct, please adjust it.

Hypoxia stimulates angiogenesis and inflammation, while systemic hypoxia can promote atherosclerosis. This occurs via a number of changes that occur, not just through increased inflammation.

Line 56: ‘was correlated with accelerate atherosclerosis’ should be accelerated

Please clarify the vein graft protocol and include a schematic. My understanding is the donor mice were placed on the 100193 diet for 3 weeks prior to surgery. It is unclear if the recipient mice were on the diet prior to the surgery and if they were on the diet after the surgery for the carbogen treatment.

Please include n values for all figures where relevant.

Fig 2A. Where are the error bars?

For the figures, Representative images and quantification do not need to be split into A and B. They should be together as they are showing exactly the same thing. E.g. Fig 3 does not need to split into 3a and b.

Line 195 should refer to figure 3 or 3B, not Fig 3A as this is only a representative image.

Figure 4 Please label the images in Fig 4 B as control and carbogen, so they are in line with the other figures and please use representative figures of the data in graph 4a. Where are the erroro bars in Figure 4A?

Figure 4 C Please include representative images indicating where the measurements were taken for vessel wall thickening.

Figure 4D Please include quantification of movat.

Figure 4E. This is the most striking difference. Why isn’t it first?

Fig 4f. Given in the images shown, the lumen are not perfectly round, is it correct to measure optimal lumen based on circumference when it would be different depending on the angle circumference measured? This measurement also provides no new information from Fig 4E.

Fig 5. What was used to measure intraplaque hemorrhage? Ter119? The stars indicating its location are hard to see and what they are indicating is very unclear. Please improve the figures.

Fig 5C. The control group has the highest number of plaques with no IPH. Are the differences between groups significant? A bar graph may be a clearer way to present this data and allow for statistical analysis and display of error bars.

Fig 6C. Please include information on light blue staining in nuclei in the legend. Please indicate area of staining 8OHdG staining in nuclei by arrow. Please include quantification.

Fig 6D and E. The figures are a bit unclear. While cleaved caspase is increased in in the carbogen treated animals, this is not significant. The fact that it associates with macrophages does not mean the macrophages themselves are apoptotic. It could be that macrophages are efferocytosing apoptotic cells to clear them as this is one of the ways they function.

Fig 7A. Not all macrophages are inflammatory and other inflammatory cells infiltrate the plaque. This Figure only looks at macrophage infiltration, please change the Figure legend accordingly. Fig 7B. Is this measured as per vessel wall area or plaque size? Because plaque size would be more correct.

Line 286. You cannot make this conclusion from the current data. Cell tracking needs to be done to assess macrophage infiltration, measuring CCL2 and TNF are not enough to draw this conclusion. Further, they are only a snapshot at a particular time point as opposed to looking at changes over time, which is required to determine infiltration. As for assuming the macrophages are apoptotic, please see comment for Figure 7A.

Line 295. How are ‘the amount of collagen and SMCs’ calculated? Do you mean SMA staining? Do you mean plaque area, rather than vessel wall?

Figure 9 These images are quite difficult to see, making it hard to tell if they are representative of the graphs.

To investigate the effects of hyperoxygenation on macrophages, surely incubating them under hyperoxic conditions would be a better way to emulate the in vivo data? Why is tBHT used?

Line 333-334 reduced macrophage content does not make a lesion more stable. It is the SMCs that contribute to plaque stability. Please correct this.
